# ATP12A Proton Pump as an Emerging Therapeutic Target in Cystic Fibrosis and Other Respiratory Diseases

**DOI:** 10.3390/biom13101455

**Published:** 2023-09-27

**Authors:** Michał Dębczyński, Giulia Gorrieri, Damian Mojsak, Floriana Guida, Federico Zara, Paolo Scudieri

**Affiliations:** 12nd Department of Lung Diseases and Tuberculosis, Medical University of Bialystok, 15-540 Bialystok, Poland; michal.debczynski@gmail.com (M.D.); damian.mojsak@gmail.com (D.M.); 2Department of Neurosciences, Rehabilitation, Ophthalmology, Genetics, Maternal and Child Health (DiNOGMI), University of Genoa, 16132 Genoa, Italy; giulia.gorrieri@edu.unige.it (G.G.); floriana.guida@edu.unige.it (F.G.); federico.zara@unige.it (F.Z.); 3Unit of Medical Genetics, IRCCS Istituto Giannina Gaslini, 16147 Genoa, Italy

**Keywords:** ATP12A, modifier genes, airway acidification, cystic fibrosis, ASL, proton transport, respiratory diseases

## Abstract

*ATP12A* encodes the catalytic subunit of the non-gastric proton pump, which is expressed in many epithelial tissues and mediates the secretion of protons in exchange for potassium ions. In the airways, ATP12A-dependent proton secretion contributes to complex mechanisms regulating the composition and properties of the fluid and mucus lining the respiratory epithelia, which are essential to maintain the airway host defense and the respiratory health. Increased expression and activity of ATP12A in combination with the loss of other balancing activities, such as the bicarbonate secretion mediated by CFTR, leads to excessive acidification of the airway surface liquid and mucus dysfunction, processes that play relevant roles in the pathogenesis of cystic fibrosis and other chronic inflammatory respiratory disorders. In this review, we summarize the findings dealing with ATP12A expression, function, and modulation in the airways, which led to the consideration of ATP12A as a potential therapeutic target for the treatment of cystic fibrosis and other airway diseases; we also highlight the current advances and gaps regarding the development of therapeutic strategies aimed at ATP12A inhibition.

## 1. Introduction

The *ATP12A* gene (also known as *ATP1AL1*) encodes the catalytic subunit of a potassium-dependent proton pump belonging to the X/K-ATPases group of the P2-type ATPase family [1,2,3]. It is similar to, but distinct from, other known ATPases: Na^+^/K^+^-ATPase (*ATP1A*) and gastric H^+^/K^+^-ATPase (*ATP4A*) [3]. ATP12A forms the non-gastric H^+^/K^+^-ATPase, secreting hydrogen ions in exchange for potassium ions in an electroneutral fashion [3].

ATP12A protein displays a wide pattern of expression in mammalian tissues. It has been found expressed in many epithelial tissues, including kidney, colon, skin, placenta, and prostate, where it is involved in acid–base and K^+^ and Na^+^ homeostasis [3,4,5,6]. It has also been observed in mammalian spermatozoa, where it may play a role in acrosome reactions [7]. Importantly, it is expressed in the respiratory tract, particularly on the apical membrane of mucus-secreting cells, both in the surface airway epithelia (goblet cells) and in the submucosal glands [8,9]. In this respect, ATP12A proton pump is one of the ion transporters regulating the composition and properties of the fluid lining the airways. In particular, it mediates potassium reabsorption and contributes to the acidification of the airway surface liquid (ASL), a process thought to be detrimental in cystic fibrosis (CF) and other chronic inflammatory respiratory diseases (Figure 1) [8,9,10,11,12].

## 2. ATP12A Proton Pump Structure and Assembly

ATP12A proton pumps exist as heterodimers composed of a catalytic α-subunit and an auxiliary β-subunit [3]. The molecular structure of the rat ATP12A was recently resolved by cryo-EM analysis [13]. The α-subunit consists of a transmembrane domain (TMD) with ten membrane-spanning helices, and N- and C-terminal cytosolic tails [13]. The TMD contains the ion-binding site residues (between transmembrane helices 4 and 8), whereas the cytosolic portion forms the actuator (A), the phosphorylation (P), and the nucleotide-binding (N) domains responsible for auto-phosphorylation and ATP hydrolysis [13]. The auxiliary β-subunit is a single-span membrane protein characterized by a large C-terminal ectodomain and a short N-terminal cytoplasmic tail [13]. The association between the α and β subunits is crucial for plasma membrane integration and targeting to the correct cell surface pole [13,14]. Interestingly, ATP12A has been proposed to associate with either the gastric β-subunit (ATP4B) or one of the β-subunits of the Na^+^/K^+^-ATPase (ATP1B1, ATP1B2 or ATP1B3), probably in a tissue-dependent manner [3]. Recent studies seem to indicate a preferential assembly of ATP12A with ATP1B1, at least in human and pig airways and mouse intestine [9,15].

## 3. ATP12A Drives Airway Acidification in CF: Lessons from Animal Models

The airways have the critical role of conducting air to and from the alveoli and acting as an active barrier to the external environment. The chemical, physical and biological features of the thin layer of fluid covering the airway epithelium, together with the properties of mucus and its clearance, modulate the airway host defense and are essential for respiratory health [16]. In CF, the reduced chloride and bicarbonate secretion caused by loss of function of the cystic fibrosis transmembrane conductance regulator (CFTR) channel disrupts the normal airway homeostasis and leads to a series of pathological alterations including ASL dehydration and acidification, sticky mucus accumulation, mucociliary transport impairment, bacteria colonization, chronic inflammation, and lung tissue damage (Figure 1) [16].

Important clues to the molecular basis for the respiratory disease CF emerged from studies on animal models, which highlighted species-specific regulatory mechanisms controlling ASL and mucus properties and emphasized the role of ASL acidification in disrupting host defense mechanisms by inhibiting the killing of bacteria and altering mucus viscosity and mucociliary clearance [11,15,17,18,19,20,21]. Importantly, such studies unveiled a pivotal role for ATP12A in mediating the ASL acidification, and thus in the pathogenesis of the CF respiratory phenotype [15]. Starting from the notion that CFTR disruption in pigs, but not in mice, produces a lung disease resembling that of humans with CF [15,17,18,19,20,21], by means of in vitro and in vivo studies, Shah and colleagues associated the divergent respiratory phenotypes observed in the three species (humans, pigs and mice) with the differential expression of ATP12A, which was found to be present at the apical membrane of human and pig airway epithelial cells but not in those of mouse airways [15]. Indeed, they found that loss of CFTR activity reduced the ASL pH in human and porcine, but not in rodent, airway epithelia [15]. ATP12A inhibition (by apical administration of ouabain) or silencing (by RNA interference) in human and porcine airway epithelia lacking CFTR was able to normalize the ASL pH, decrease the mucus viscosity, and increase the antibacterial activity [15]. As further evidence that ATP12A-mediated ASL acidification is a crucial driver of CF lung pathogenesis, the induction of ATP12A expression in mouse epithelia resulted in ASL acidification, increased mucus viscosity, and impaired bacterial killing, effects which could be reverted by ouabain [15]. Such findings were also confirmed in vivo, with CF mice overexpressing ATP12A which developed evidence of inflammation and were more prone to bacterial infections, which is similar to what has been observed in untreated CF pigs [15]. Therefore, these studies explained the molecular basis of the different CF lung phenotypes observed in humans, pigs, and mice, and evidenced for the first time a detrimental role for ATP12A proton pumping in the pathogenesis of CF lung disease [15].

## 4. ATP12A Expression, Function, and Modulation in CF and Other Respiratory Diseases

The proper composition and properties of the liquid and mucus covering the airways are finely tuned by a series of mechanisms in which the activity of different ion channels and transporters plays a crucial role [16,22]. Therefore, it is not surprising that alterations in the expression and/or function of one or more of these actors may play relevant roles in the pathogenesis of CF and other respiratory disorders, such as allergic airway diseases and chronic obstructive pulmonary disease (COPD) [16,23,24].

Coakley and colleagues first reported ATP12A expression in the lumen of human bronchial airways and its role in the acidification of ASL in cultured bronchial epithelia from CF and non-CF individuals [10]. As confirmed by many, but not all, subsequent studies, they found a more acidic pH in CF ASL compared to healthy ones, despite a similar rate of ATP12A-mediated proton pumping [9,10,15,19,20,25]. This finding suggests that, under physiological conditions, ATP12A-dependent proton secretion may act in concert with CFTR-mediated bicarbonate transport to preserve the right acid–base balance in the ASL (Figure 1, top cartoon). In CF, defective CFTR reduces the transepithelial chloride and bicarbonate secretion leaving the action of ATP12A unbalanced, leading to the acidification of ASL (Figure 1, bottom cartoon) [9,10,15,26,27,28]. Such an acidic ASL may contribute to CF disease pathogenesis by impairing mucus properties and antimicrobial defenses on the airway surface [15,16,19,20,21,22].

More recently, studies by Galietta’s lab strengthened the association between ATP12A and CF by showing that ATP12A expression is markedly increased in the airways of CF patients [9,29]. In a first work, they assessed ATP12A protein expression in bronchial biopsies collected from CF and non-CF patients subjected to lung transplantation, and, therefore, at a late stage of the disease [9]. ATP12A appeared upregulated in the CF cohort and was mainly localized on the apical membrane of goblet cells in the surface epithelium and in submucosal glands [9]. In a second work, they enrolled a larger number of patients and collected upper airway epithelial cells through a nasal brushing procedure; these cells were freshly fixed and subjected to immunofluorescence detection of ATP12A protein as previously done in sections of bronchi [29]. Although showing a variable extent of expression, the increased detection of ATP12A in CF was confirmed and, importantly, observed also in very young patients, suggesting that ATP12A upregulation is an early event during CF disease progression [29]. In the era of highly effective CFTR modulator therapies, it would be relevant to address, in future studies, whether such therapeutic treatments exert some effects on the expression of ATP12A in the airways of CF patients.

Interestingly, ATP12A overexpression did not occur in bronchial epithelial cells collected from CF donors and cultured in vitro under control and sterile conditions, indicating that the overexpression found in vivo is not a direct consequence of the CF genotype [9,30]. On the other hand, ATP12A overexpression was also found in bronchial airways of a non-CF patient with IgG deficiency who suffered from frequent pulmonary bacterial infections, and in nasal cells from non-CF individuals with rhinitis [9,29]. Moreover, a recent study by Abdelgied and colleagues showed ATP12A upregulation in the distal small airways of idiopathic pulmonary fibrosis (IPF) and in submucosal glands and large airways of both IPF and COPD lung tissues collected from human donors [31]. These observations revealed that ATP12A upregulation is not specific to CF but happens under various pathological conditions, possibly because of infectious and inflammatory stimuli.

Many in vitro studies support this hypothesis, showing a marked induction of ATP12A expression after chronic exposure of cultured airway epithelial cells to bacterial components or inflammatory cytokines such as IL-4, IL-13, and IL-17 [8,9,23,29]. Importantly, ATP12A upregulation under these conditions is not an isolated event but appears as part of complex processes involving the modulation of many genes coding for ion channels and transporters, including, for example, the SLC26A4 chloride/bicarbonate exchanger, the TMEM16A chloride channel, the SLC12A2 NKCC1 co-transporter, the ENaC sodium channel, and the KCNJ16 and KCNK3 potassium channels [8,29]. Among others, SLC26A4, which plays a relevant role in bicarbonate secretion in the airways, is often upregulated in parallel with ATP12A, further highlighting how the fine and coordinated tuning of acid–base secretion may be important for several aspects of airway homeostasis, such as, for example, mucus dynamics [8,10,15,19,20,29].

In this regard, Lennox and colleagues started dissecting the contribution of ATP12A to mucus dysfunction produced by IL-13, a key mediator of Type-2 airway inflammation [23]. Primary human bronchial epithelial cells cultured on an air–liquid interface and exposed to IL-13 showed higher liquid absorption and proton secretion rates, which resulted in decreased ASL volume, impaired mucociliary function (evidenced by the reduction of the fraction of functional ciliated airway and of the rate of rotational mucociliary transport), and markedly increased ASL viscosity [23]. ATP12A expression was increased three-fold after IL-13 treatment [23]. Lentiviral delivery of shRNA targeting ATP12A decreased ATP12A protein expression under both control and IL-13 treatment conditions and reduced the proton secretion rate without, however, affecting the net ASL pH [23]. Importantly, despite the small effect on the steady-state ASL pH, ATP12A silencing resulted in normalization of ASL volume and viscosity [23]. These findings identified ATP12A as the main driver of the detrimental rheological changes produced during Type-2 airway inflammation and suggest that ATP12A may also promote mucus alteration by pH-independent mechanisms, such as potassium absorption or mucin modification, as speculated by the authors [23]. 

An example of airway disease associated with an increased level of IL-13 is chronic rhinosinusitis with nasal polyps (CRSwNP), which is characterized by local inflammation of the sinonasal mucosa and tissue eosinophilia [24]. Interestingly, Min and colleagues, found that ATP12A silencing in nasal epithelial cells attenuated IL-13-induced expression of the eosinophil-specific chemoattractant eotaxin-3, possibly highlighting an additional mechanism through which ATP12A may exert its function [24]. Moreover, they found that similar attenuation could be obtained with different inhibitors of the gastric proton pump, including SCH-28080 and the clinically approved proton pump inhibitors (PPIs) omeprazole, lansoprazole, rabeprazole, pantoprazole, and esomeprazole, suggesting that these compounds could also be effective on the ATP12A proton pump [24]. Although such effects could be explained by the relatively high (65%) sequence identity between ATP4A (the gastric proton pump) and ATP12A, conclusive evidence of a direct inhibitory effect of PPIs on ATP12A is still lacking, and alternative mechanisms of action can be hypothesized. For example, Delpiano and colleagues found that esomeprazole acts by reducing ATP12A expression [30]. Specifically, they found that esomeprazole exerted variable effects on the ASL pH of cultured CF bronchial epithelia: while acute administration paradoxically acidified ASL in an ATP12A-independent manner, chronic exposure resulted in ASL alkalinization, without any deleterious effects on the epithelia, due to decreased mRNA expression of ATP12A [30]. As a second example, Xiong and colleagues reported that lansoprazole reduced the frequency of acute exacerbation of COPD [32]. In this case, the authors observed a lowering of the levels of inflammatory cytokines (IL-1β, IL-6, IL-8, TNF-α, and GM-CSF) in the sputum of COPD patients treated with the PPI. However, no evidence of a direct connection with ATP12A function or expression was proved [32].

Besides ouabain, SCH-28080, and PPIs, vonoprazan, a potassium-competitive proton pump blocker, was also reported as a potential tool to inhibit ATP12A [31]. Abdelgied and colleagues, found that vonoprazan, like ouabain, increased the ASL pH of IPF small airway models and reduced bleomycin-induced pulmonary fibrosis in mice with ATP12A overexpression (produced by intratracheal instillation of adenovirus encoding ATP12A) [31].

## 5. *ATP12A* as a Modifier of Meconium Ileus in CF 

CF also affects the intestine and pancreas (meconium ileus, malabsorption, diabetes and pancreatic insufficiency), liver (biliary cirrhosis), vas deferens (infertility), and sweat glands (heat shock) [33]. Moreover, CF shows significant phenotypic variability, as a wide spectrum of disease severity is also reported in patients with the same genotype, suggesting potential important roles for gene modifiers that may influence the clinical expression of the disease [34,35,36]. Surprisingly, in addition to its relevance in CF airways, *ATP12A* has also been identified as a susceptibility locus for meconium ileus risk by the largest study population of individuals with CF from the international CF Gene Modifier Consortium (GMC) [37]. Meconium ileus is one of the earliest and most severe manifestations of CF, present in up to approximately 20% of CF newborns, and it is due to intestinal and pancreatic dysfunctions causing obstruction of the small intestine by inspissated meconium [38].

To investigate the potential mechanism of *ATP12A* association with meconium ileus, Gong and colleagues developed a colocalization framework that integrates transcriptomic and genetic association information [37]. Interestingly, meconium ileus risk was associated with increased ATP12A expression in the pancreas [37]. The regulation of proton and bicarbonate movements plays important role in fluid secretion in the pancreatic ducts, as maintaining appropriate pH enables proper enzyme transport and prevents auto-activation [33,39,40]. In the absence of functional CFTR, a progressive acidification of the acinar and duct lumen leads to defective secretion of the pancreatic juice and lumen obstruction [33,39,40,41]. As observed in the airways, it can be hypothesized that ATP12A overexpression in the pancreas may contribute the excessive fluid acidification and thus worsen the CF pancreatic dysfunction. Interestingly, ATP12A was found to be expressed in human pancreatic duct in in vitro models, as a monolayer formed by the Capan-1 cell line, which may represent a useful tool to better investigate the role of this proton pump in the pancreas [42].

## 6. Potential Strategies for ATP12A Targeting

Recent studies on ATP12A have opened new perspectives about its role in respiratory physiology and diseases and have paved the way for the future development of new therapeutic approaches aimed at targeting its function and/or expression. Indeed, as evidenced in earlier paragraphs there are many associative, mechanistic, and also pharmacological clues, obtained both in vivo and in vitro, supporting the idea that ATP12A inhibition could represent a potential treatment for CF and also other respiratory diseases, including IPF, COPD, CRSwNP, and other inflammatory airway diseases: (i) ATP12A appears overexpressed in different diseases characterized by airway inflammation and/or infection [9,24,29,31,37]; (ii) ATP12A overexpression produces negative effects in the airways, such as ASL acidification, mucus alterations and tissue fibrosis [9,10,15,24,31]; and (iii) ATP12A ablation/inhibition was efficacious in reverting/correcting the detrimental effects produced by its upregulation/function [9,10,15,23,24,30,31].

A therapeutic approach based on ATP12A inhibition can be particularly useful in CF, where not all patients can benefit from the highly effective CFTR modulator therapies [16]. Indeed, developing therapeutics also for the modulation of alternative targets, and not only devoted to CFTR rescue, could be essential for CF patients expressing undruggable CFTR mutants and could also be useful as an adjuvant therapy supporting the effect of CFTR modulators.

For this purpose, different therapeutic strategies based on ATP12A targeting can be envisioned, as outlined in Figure 2.

The pharmacological modulation of ion channels and transporters has been proved to be a good therapeutic approach for different diseases, including CF, with the recent advent of highly effective CFTR modulator therapies [16]. However, in the case of ATP12A, effective, potent, and specific inhibitors are completely lacking to date. The cardiac glycoside ouabain has been extensively used at very high concentrations (100–200 μM) to inhibit ATP12A in in vitro experiments [9,10,15,29,30,43]. However, this compound is obviously not an ideal candidate drug for targeting ATP12A given its strong effect on the basolateral Na^+^/K^+^-ATPase (which is sensitive to sub-micromolar ouabain concentrations) and its resulting adverse effects in vivo [44]. Besides ouabain, other small molecules, also comprising already approved drugs, are under investigation for their potential use as ATP12A inhibitors: SCH-28080, vonoprazan, and different PPIs [24,30,31,32]. These are all known inhibitors of the gastric proton pump, although with different chemical structures and mechanisms of action [45]. The rationale for their testing is based mainly on the relatively high sequence identity between their classic target (ATP4A) and ATP12A, presenting the opportunity for a drug reposition approach. However, although few promising and preliminary results indicate that some of these compounds may exert some effects on ATP12A, their specificity and mechanisms of action (e.g., direct inhibition of the proton pumping activity or ATP12A mRNA degradation) should be clarified in future studies. In addition, their action should be assessed in appropriate airway models and conditions, as their effectiveness could be affected by the particular environment in which the ATP12A proton pump works. For example, the effect of compounds acting as potassium-competitive blockers could be hindered by the relatively high potassium concentration found in the ASL [46,47].

A parallel strategy may involve the search for ATP12A inhibitors with new chemical scaffolds by undertaking the screening of large chemical libraries. In this respect, validated cell models and assays to specifically interrogate ATP12A function and to be used in high-throughput applications are needed.

A second strategy to achieve ATP12A inhibition may involve the suppression of its expression rather than its function. The rationale of this approach may be found in the deleterious overexpression of ATP12A observed in CF and other airway inflammatory diseases [9,24,29,31,37]. Importantly, proofs of concept of this potential approach have been obtained in many studies by either RNA interference (through viral delivery of shRNA directed against ATP12A mRNA) or genome editing (CRISPR/Cas9-mediated knockdown or ablation of *ATP12A* gene expression) [15,23,24,30]. However, despite the great potential and the enormous advances made in RNA-based and CRISPR/Cas9-based therapeutic strategies, there are still several issues to overcome, particularly in the field of airway diseases. Among these are the following: being able to find an efficient transgene vector that does not create strong immunogenicity and must be able to penetrate the airways mucus layer, targeting the specific cell type in the complex respiratory system, and maintaining an effect over time that does not cause toxicity [48,49,50]. To date, many trials using RNA-based and DNA-based therapeutics have been done or are in progress. The use of recombinant adeno-associated virus (rAAV) as a vector seems to be the most explored method of delivery [50,51,52,53], but other nanodelivery tools, such as lipid nanoparticle encapsulating Cas9 mRNA and sgRNA, are also under investigation [54,55,56].

Finally, an additional and putative way to obtain ATP12A inhibition could involve interference with its trafficking to the plasma membrane. Indeed, it has been shown in different studies, and by using different cell models, that ATP12A requires assembly with an auxiliary β-subunit to reach the plasma membrane and function as a proton pump [3,9,13,14]. Accordingly, searching for tools, such as small molecules or peptides, that are able to alter the interaction between them, and thus prevent their assembly and trafficking to the plasma membrane, could be proposed as a new strategy to block ATP12A-dependent proton secretion. In this regard, the resolution of the molecular structure of the human ATP12A and the identification of domains involved in interaction and assembly with the auxiliary subunit would also pave the way for structure-based virtual screening and molecular docking approaches [57].

## Figures and Tables

**Figure 1 biomolecules-13-01455-f001:**
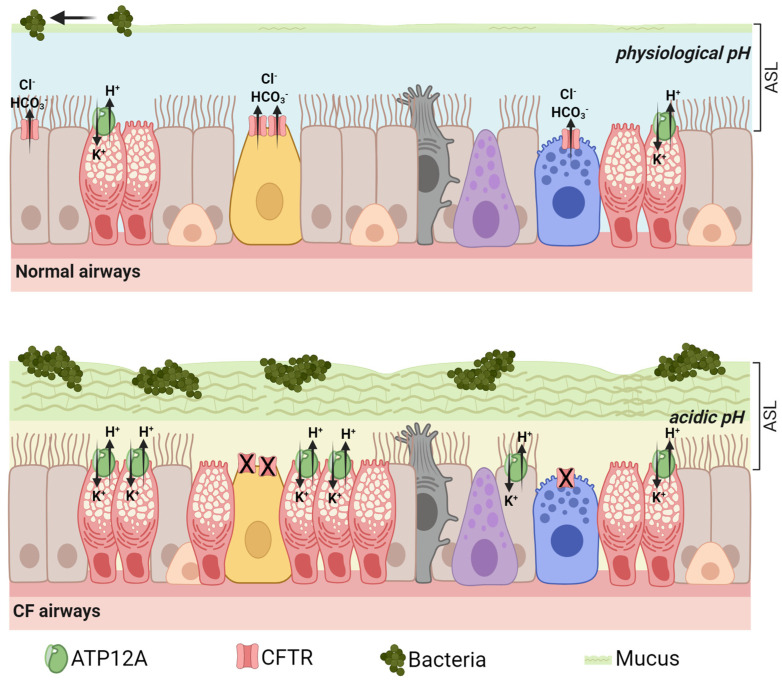
ATP12A expression in normal vs. CF airways. In normal airways (**top**), ion channels and transporters work in concert to mediate the transepithelial absorption and secretion of Cl^−^, HCO_3_^−^, and H^+^ as well as those of other solutes, which in turn regulate the chemical and physical characteristics of ASL. The homeostasis of ASL guarantees the proper function of mucociliary clearance (MCC). In CF (**bottom**), the less hydrated airways suffer the reduction of MCC, and a severe inflammatory process triggers subsequent changes: goblet cell hyperplasia, increased mucus secretion, ASL acidification due to the increased presence and unbalanced function of *ATP12A*, and alteration of ASL and mucus properties leading to mucus stasis.

**Figure 2 biomolecules-13-01455-f002:**
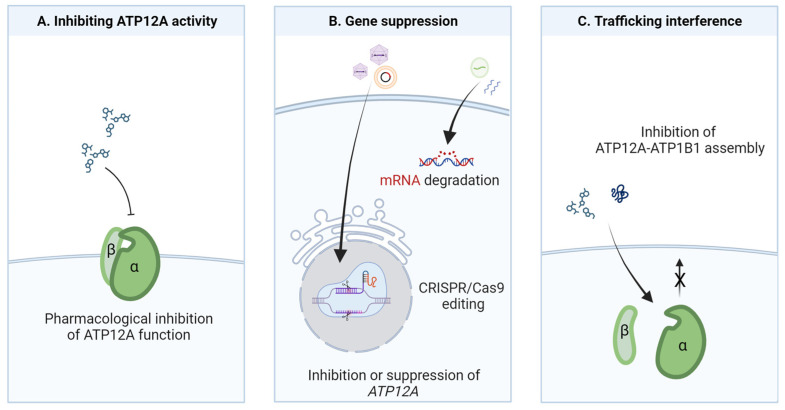
Potential strategies for ATP12A targeting in airway diseases. (**A**) Inhibition of ATP12A proton pump function can be achieved by screening libraries of small molecules with appropriate functional assays. (**B**) *ATP12A* gene suppression or downregulation can be achieved by targeting genomic DNA (genome editing) or mRNA (antisense strategy) through different delivery agents. (**C**) Trafficking interference could be achieved by in silico screening of chemical libraries to find molecules hindering α/β subunit assembly and therefore the trafficking of the proton pump to the plasma membrane.

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
