# Peer review of "ATP12A Proton Pump as an Emerging Therapeutic Target in Cystic Fibrosis and Other Respiratory Diseases"

_biomolecules, 2023, doi:10.3390/biom13101455_

Round 1

Reviewer 1 Report

I found this a well written review that is easy to read and follow.

The topic is relevant and this provides a nice summary of knowledge to date re. ATP12A in the airway, physiological role and potential therapeutic target.

It has already been type set and is well presented.

The only point I would make is - is it known what effect treatment with a highly-effective CFTR modulator has on ATP12A expression and level of function in respiratory epithelia in vitro and in vivo? If proposed as an adjunctive treatment this is important to know. 

Author Response

We thank the reviewer for her/his very positive comments, and also to rising the point about the effect of highly-effective CFTR modulator therapy on ATP12A expression. To the best of our knowledge there are no published studies investigating this very important issue. We added a sentence in the review (lines 141-143) to highlight this need.

Reviewer 2 Report

The authors provide a concise review describing the role of ATP12A in airway disease and its role in acidification. The review is timely with recent additions to the pulmonary ATP12A literature and from a group with the appropriate expertise. The reviewer has several suggestions, which are mostly minor.

Major:

Figure 1: It is hard to appreciate ion flux as drawn. One possibility is to add arrows. The authors could also consider reducing the number of floating ions and simply using “normal pH” and “acidic”. Further, the second panel shows more K+ inside the cell; however, people with CF and the porcine CF model do not have a reduction in ASL K+. As the authors state in the text and figure legend, a driver of ASL acidification is a lower concentration of HCO3, thereby HCO3 buffering capacity, in the ASL, but Figure 1 looks like the acidification is driven by an increase in ATP12A turnover. This could be separated from the effects of inflammation by adding a third panel “CF airways + Inflammation” showing an increase in goblet cells and ATP12A expression. If using the 3 panel approach, the current “CF airways” should have the same amount of goblet cells.

At least a paragraph must address that base secretion (e.g., by pendrin) often increases in parallel with ATP12A.

Minor:

Line 87: For the finding that 2 species acidify and have hallmarks of CF vs. another that does not acidify ASL and does not have hallmarks, the term “correlated” seems too quantitative, especially because the mechanisms of pH regulation can be complex. For example, expression of pendrin induced by cytokines can alkalinize the ASL to a similar degree as ATP12A inhibition. The subsequent sentence “ATP12A inhibition or silencing in human…” appropriately summarizes these findings. The reviewer suggests that the sentence “Intriguingly,…” be removed and to omit the word “moreover,” in the following sentence.

Line 38: K+/Na+ is formatted similarly to a cotransporter; writing “K+ and Na+ homeostasis” would be clearer.

Lines 270-271. The authors should also note that the Na/K pump has a higher IC50 for ouabain than ATP12A (e.g., 10.1152/ajprenal.1996.271.3.F539) within this sentence.

Among Lines 273-276. It should be noted that an obstacle for ATP12A antagonists that compete with K+ is that the ASL has a relatively high [K+] (10.1172/JCI119802 and 10.1074/jbc.M808021200).

Figure 2: Pharmacological inhibition covers both the left and right panels. Panel A could be renamed to "Inhibiting ATP12A Activity” or “Inhibiting ATP12A Turnover”.

Although the authors may choose not to address this comment, some points that the reviewer has not seen in the literature include:

1)    Why do we need ATP12A to remove acid from airway cells?

2)    Why do airways move acid into the ASL vs. moving acid into the interstitium?

-NHE1 in the basolateral membrane and its pHi dependence ought to be enough to maintain pHi and protect the ASL from becoming acidic.

Author Response

REVIEWER 2

The authors provide a concise review describing the role of ATP12A in airway disease and its role in acidification. The review is timely with recent additions to the pulmonary ATP12A literature and from a group with the appropriate expertise. The reviewer has several suggestions, which are mostly minor.

Major:

Figure 1: It is hard to appreciate ion flux as drawn. One possibility is to add arrows. The authors could also consider reducing the number of floating ions and simply using “normal pH” and “acidic”. Further, the second panel shows more K+ inside the cell; however, people with CF and the porcine CF model do not have a reduction in ASL K+. As the authors state in the text and figure legend, a driver of ASL acidification is a lower concentration of HCO3, thereby HCO3 buffering capacity, in the ASL, but Figure 1 looks like the acidification is driven by an increase in ATP12A turnover. This could be separated from the effects of inflammation by adding a third panel “CF airways + Inflammation” showing an increase in goblet cells and ATP12A expression. If using the 3 panel approach, the current “CF airways” should have the same amount of goblet cells.

We thank the reviewer for her/his useful comments and suggestions on the Figure 1. We have modified the figure by including arrows to indicate the ion fluxes and by reducing the number of floating ions. We have also corrected the point about the potassium transport.

At least a paragraph must address that base secretion (e.g., by pendrin) often increases in parallel with ATP12A.

We thank the reviewer for this suggestion. We added a short paragraph reporting that “…ATP12A upregulation… is not an isolate event but appears as part of complex processes involving the modulation of many genes coding for ion channels and transporters, including for example the SLC26A4 chloride/bicarbonate exchanger, the TMEM16A chloride channel, the SLC12A2 NKCC1 co-transporter, the ENaC sodium channel, and the KCNJ16 and KCNK3 potassium channels [8,29]. Among others, SLC26A4, which play a relevant role in bicarbonate secretion in the airways, is often up-regulated in parallel with ATP12A, further highlighting how the fine and coordinated tuning of acid-base secretion may be important for several aspects of the airway homeostasis, as for example the mucus dynamics…” 

Minor:

Line 87: For the finding that 2 species acidify and have hallmarks of CF vs. another that does not acidify ASL and does not have hallmarks, the term “correlated” seems too quantitative, especially because the mechanisms of pH regulation can be complex. For example, expression of pendrin induced by cytokines can alkalinize the ASL to a similar degree as ATP12A inhibition. The subsequent sentence “ATP12A inhibition or silencing in human…” appropriately summarizes these findings. The reviewer suggests that the sentence “Intriguingly,…” be removed and to omit the word “moreover,” in the following sentence.

Done

Line 38: K+/Na+ is formatted similarly to a cotransporter; writing “K+ and Na+ homeostasis” would be clearer.

Done

Lines 270-271. The authors should also note that the Na/K pump has a higher IC50 for ouabain than ATP12A (e.g., 10.1152/ajprenal.1996.271.3.F539) within this sentence.

Done, new line 278

Among Lines 273-276. It should be noted that an obstacle for ATP12A antagonists that compete with K+ is that the ASL has a relatively high [K+] (10.1172/JCI119802 and 10.1074/jbc.M808021200).

Done, new lines 289-293: “Also, their action should be assessed in appropriate airway models and conditions, as their effectiveness could be affected by the particular environment in which ATP12A proton pump works. For example, the effect of compounds acting as potassium-competitive blockers could be hindered by the relatively high potassium concentration found in the ASL”

Figure 2: Pharmacological inhibition covers both the left and right panels. Panel A could be renamed to "Inhibiting ATP12A Activity” or “Inhibiting ATP12A Turnover”.

Done

Although the authors may choose not to address this comment, some points that the reviewer has not seen in the literature include:

1)    Why do we need ATP12A to remove acid from airway cells?

2)    Why do airways move acid into the ASL vs. moving acid into the interstitium?

-NHE1 in the basolateral membrane and its pHi dependence ought to be enough to maintain pHi and protect the ASL from becoming acidic.

We thank the reviewer for this final comment and perfectly agree with her/him about the relevance of the points raised. Indeed, both questions should be solved by future studies further investigating the role of ATP12A and other proton transporters, also considering the potential importance of ATP12A in mediating potassium absorption besides proton secretion.